# *MTHFR* c.665C>T and c.1298A>C Polymorphisms in Tailoring Personalized Anti-TNF-α Therapy for Rheumatoid Arthritis

**DOI:** 10.3390/ijms24044110

**Published:** 2023-02-18

**Authors:** Amin Ravaei, Lia Pulsatelli, Elisa Assirelli, Jacopo Ciaffi, Riccardo Meliconi, Carlo Salvarani, Marcello Govoni, Michele Rubini

**Affiliations:** 1Medical Genetics Laboratory, Department of Neuroscience and Rehabilitation, University of Ferrara, 44121 Ferrara, Italy; 2Laboratory of Immunorheumatology and Tissue Regeneration, IRCCS Istituto Ortopedico Rizzoli, 40136 Bologna, Italy; 3Medicine and Rheumatology Unit, IRCCS Istituto Ortopedico Rizzoli, 40136 Bologna, Italy; 4Division of Rheumatology, Azienda USL-IRCCS di Reggio Emilia, 42122 Reggio Emilia, Italy; 5University-Hospital of Modena, University of Modena and Reggio Emilia, 41124 Modena, Italy; 6Section of Hematology and Rheumatology, Department of Medical Sciences, University of Ferrara, 44121 Ferrara, Italy; 7Rheumatology Unit, Sant’Anna University Hospital, 44124 Ferrara, Italy; 8University Center for Studies on Gender Medicine, University of Ferrara, 44121 Ferrara, Italy

**Keywords:** rheumatoid arthritis, TNF-α inhibitors, genetic association, *MTHFR*, biomarkers, pharmacogenetics, personalized medicine

## Abstract

Rheumatoid arthritis (RA) is an inflammatory autoimmune disease with a prevalence of 1%. Currently, RA treatment aims to achieve low disease activity or remission. Failure to achieve this goal causes disease progression with a poor prognosis. When treatment with first-line drugs fails, treatment with tumor necrosis factor-α (TNF-α) inhibitors may be prescribed to which many patients do not respond adequately, making the identification of response markers urgent. This study investigated the association of two RA-related genetic polymorphisms, c.665C>T (historically referred to as C677T) and c.1298A>C, in the *MTHFR* gene as response markers to an anti-TNF-α therapy. A total of 81 patients were enrolled, 60% of whom responded to the therapy. Analyses showed that both polymorphisms were associated with a response to therapy in an allele dose-dependent manner. The association for c.665C>T was significant for a rare genotype (*p* = 0.01). However, the observed opposite trend of association for c.1298A>C was not significant. An analysis revealed that c.1298A>C, unlike c.665C>T, was also significantly associated with the drug type (*p* = 0.032). Our preliminary results showed that the genetic polymorphisms in the *MTHFR* gene were associated with a response to anti-TNF-α therapy, with a potential significance for the anti-TNF-α drug type. This evidence suggests a role for one-carbon metabolism in anti-TNF-α drug efficacy and contributes to further personalized RA interventions.

## 1. Introduction

Rheumatoid arthritis (RA) is an autoimmune and chronic inflammatory disease that symmetrically affects the polyarthritis of small and large joints, and causes joint and periarticular structural damage [1]. RA has a prevalence of 0.4% to 1.3% across the world population [2] and induces significant morbidity with a decreased quality of life and an increased mortality rate [3]. The symptoms of RA vary between early and advanced RA. Early RA—characterized by symptoms such as a flu-like feeling, fatigue, morning stiffness, and swollen and tender joints, which are accompanied by increased levels of C-reactive protein (CRP) and an increased erythrocyte sedimentation rate (ESR) [4]—differs from advanced and insufficiently treated RA, which is characterized by severe manifestations such as pleural effusions, lung disease, vasculitis, lymphomas, atherosclerosis, hematologic abnormalities, keratoconjunctivitis, rheumatic nodules, joint malalignments, motion limitations, and bone and cartilage destruction (reviewed in detail in [5,6,7]).

According to the American College of Rheumatology (ACR) and the European Alliance of Associations for Rheumatology (EULAR), the current therapeutic goal for RA treatment is to achieve remission or a low disease activity [8,9] by following a structured algorithm of add-on and switch-off disease-modifying anti-rheumatic drug (DMARD) therapies [10]. Generally, the treatment is initiated by one of the conventional synthetic DMARDs such as methotrexate, sulfasalazine, hydroxychloroquine, or leflunomide as a monotherapy or a combination therapy [11]. DMARD efficacy varies significantly among RA patients. For example, methotrexate, which is usually prescribed as the first-line drug, is not effective for 30% to 50% of patients [12,13,14,15,16]. If the response to the first-line therapy is not adequate or fails within 3 to 6 months of the initiation, tumor necrosis factor-α (TNF-α) inhibitors, as biologic DMARDs, should be added to the therapy. Moreover, according to the ACR, if early RA patients experience a high disease activity followed by poor prognosis factors, the use of TNF-α inhibitors as an immediate first-line therapy is recommended [17].

Currently, there are five TNF-α inhibitor drugs, which present similar efficacy and safety profiles. These drugs include adalimumab, golimumab, infliximab, certolizumab, and etanercept [17]. Although these TNF-α inhibitor drugs have significantly improved RA treatment, approximately 40% to 44% of patients do not respond to them adequately [18] and might present with complications such as the development of serious adverse effects, including severe infections, malignancies, congestive heart failure, demyelinating disorders, skin reactions, and drug-induced lupus [19] or a reduced efficacy of the therapy due to the immunogenicity of the drug [20]. Therefore, the identification of the response markers for these drugs would provide optimal treatments for both responder and non-responder patients. Efficient pharmacogenetic markers, with the capability of promptly identifying patients with a lower chance to respond to therapy, could facilitate the provision of individually tailored therapies and ultimately prevent the progression of the disease. So far, several single nucleotide polymorphisms (SNPs) in different known loci such as *NUBPL* (rs2378945), *NCTN5* (rs1813443), *PLA2G4A* (rs12142623 and rs4651370) [21], *LINC02549* (rs7767069), *LARRC55* (rs717117G) [22], *MED15* (rs113878252), *MAFB* (rs6065221) [23], *CD84* (rs6427528 and rs1503860) [24], *TNF* (rs1800629), *EYA4* (rs17301249) [25], *PDZD2* (rs1532269) [26], and *CCL21* (rs2812378) [27] genes or in unknown loci, including rs4411591, rs7767069, rs1447722, and rs1568885 [21], have been identified, which are associated with a response to anti-TNF treatments in RA. The evidence supports an association between SNPs in the *MTHFR* genes c.665C>T (rs1801133, historically referred to as c.677C>T or C677T) and c.1298A>C (rs1801131) and the occurrence risk of RA [28,29,30] or the expression of inflammation markers [31,32], conditions during which inflammatory cytokines such as TNF-α, which is the direct target of TNF-α inhibitor drugs, play a fundamental role [33]. However, further studies addressing the association of these polymorphisms with the response to TNF-α inhibitor drugs in RA patients have not been performed. Therefore, in this study, we aimed to investigate the association of these two SNPs with the response to TNF-α inhibitor drugs in RA patients.

## 2. Results

### 2.1. Characteristics of Study Subjects and Response to Therapy

After the follow-up period, 81 patients who received anti-TNF-α therapy as a monotherapy or in combination with other phase I drugs (including hydroxychloroquine, corticosteroids, MTX, sulfasalazine, and leflunomide) were selected and included in the further analyses. Patients who responded successfully to the first-line therapy, or who received non-TNF-α-targeting biologic DMARDs, or who did not complete the follow-up period, or with recorded data not passing the quality control were excluded. The average age of these patients was 56.24 ± 12.96 years and 67.9% of them were female. Among the patients, 4 did not respond to the first anti-TNF-α drug choice and received a second anti-TNF-α drug, which increased the record size to 85. Overall, 60% of the patients responded to therapy with an anti-TNF-α drug, and there was no significant difference between the monotherapy or combination therapy groups either with the overall response (GR+MR vs. NR) to the treatment (*p* = 0.30) or with the degree of response (GR vs. MR, *p* = 0.46; GR vs. NR, *p* = 0.98; MR vs. NR, *p* = 0.12). The characteristics of the patients are summarized in Table 1.

Among the studied parameters, no associations were detected between the response to the anti-TNF-α therapy and parameters such as sex (*p* = 0.79), age (*p* = 0.21), exposure to tobacco smoking (*p* = 0.78), or the presence of RF in serum (*p* = 0.52) or ACPA (*p* = 0.70).

### 2.2. SNP Association Analysis

The genotype distribution of both the c.665C>T and c.1298A>C SNPs were in a Hardy–Weinberg equilibrium; their frequencies are presented in Table 2.

The analyses showed that the c.665C>T genotypes were associated with the response to the anti-TNF-α therapy in an allele dose-dependent manner. The association was significant for the TT homozygous patients, presenting a 7-fold increased chance of responding to anti-TNF-α therapy compared with CC homozygotes (OR = 7.00, 95% CI 1.60–30.80, *p* = 0.010). The RA cases bearing a CT heterozygous genotype showed a moderate trend towards an association with the response to therapy (OR = 2.14, 95% CI 0.78–5.91, *p* = 0.14). Regarding the *MTHFR* c.1298A>C polymorphism, a similar but reversed allele dose-dependent trend of an association with a response to the anti-TNF-α therapy was observed. Among the responders, the AC heterozygous patients showed a trend towards a reverse association with a response to the therapy with TNF-α inhibitors (OR = 0.52, 95% CI 0.21–1.32, *p* = 0.17) whereas the CC homozygous cases showed a 12-fold reduced chance of responding to therapy compared with AA homozygotes (OR = 0.082, 95% CI 0.009–0.78, *p* = 0.029) (Figure 1).

Further analyses among the responders revealed that the association with c.665C>T was stronger in the GR group (CT heterozygous: OR = 3.02, 95% CI 0.70–13.00, *p* = 0.14; TT homozygous: OR = 10.89, 95% CI 1.73–68.54, *p* = 0.01) compared with the MR group (CT heterozygous: OR = 1.77, 95% CI 0.56–5.53, *p* = 0.34; TT homozygous: OR = 5.33, 95% CI 1.07–26.61, *p* = 0.04). Similarly, the level of response showed a very comparable association with c.1298A>C among the heterozygous subjects (GR: OR = 0.56, 95% CI 0.18–1.73, *p* = 0.32; MR: OR = 0.50, 95% CI 0.17–1.42, *p* = 0.19). However, due to the small sample size, it was not possible to compare the level of response in the CC homozygous subjects (Figure 2).

As both *MTHFR* polymorphisms presented an association with a response to the therapy with anti-TNF-α drugs, we wondered whether the association was influenced by the specific type of inhibitor. As shown in Figure 3, a similar allele dose-dependent trend towards an association of c.665C>T genotypes with the therapy response was observed among the RA cases treated with anti-TNF-α monoclonal antibody (mAb) drugs and in cases treated with an anti-TNF-α fusion protein (FP). However, the stratification of cases according to the type of TNF-α inhibitors provided evidence that among patients with the c.1298AC genotype, the trend towards a reduced response to therapy was restricted only to cases that had been treated with anti-TNF-α mAbs (OR = 0.15, 95% CI 0.03–0.71, *p* = 0.017). No AC genotype effect was observed in those treated with TNF-α FP (OR = 1.29, 95% CI 0.38–4.39, *p* = 0.69) (Figure 3). Due to the lack of CC homozygous cases treated with anti-TNF-α FP, it was not possible to determine the influence of the drug type among the CC homozygotes.

## 3. Discussion

In the present study, we have provided evidence that in patients with RA, the response to anti-TNF-α therapy is influenced by common polymorphisms of the *MTHFR* gene in an allele dose-dependent manner. Both c.665C>T and c.1298A>C variants have been reported to be associated with RA occurrence [29,30] but, to our knowledge, this is the first evidence of an association between the *MTHFR* variants and a response to anti-TNF-α drugs.

Notably, the minor allele of the c.665C>T variant was associated with a therapeutic response whereas the c.1298A>C variant was inversely associated. The finding of an opposite effect of the two studied variants was not surprising because of their close physical proximity and the consequent high linkage disequilibrium between them as well as the fact that their minor alleles are very rarely detected on the same chromosome [34].

The MTHFR enzyme catalyzes the synthesis of 5-methyltetrahydrofolate and contributes to the removal of homocysteine by its remethylation to methionine [35]. The C>T substitution at nt 665 determines an alanine-to-valine change that increases the thermolability of the enzyme and impairs the binding of flavin adenine dinucleotide (FAD) and, therefore, causes a reduced catalytic activity [36]. Individuals carrying the c.665TT genotype have an increased need for folate intake in order to maintain adequate concentrations of folate in the serum and red blood cells as well as to avoid increasing the level of total homocysteine (tHcy) in their plasma [37].

The A-to-C transition at nt 1298 results in a glutamate-to-alanine substitution in the regulatory-binding domain of the enzyme, causing the reduced binding of S-adenosyl-methionine (SAM) and leading to a decreased enzyme activity [38], but to a lesser extent than the c.665C>T variant [39].

Both variants are associated with a higher expression of inflammation markers [31,32] and the risk of occurrence of RA [28,29,30]. Although the exact mechanism of the association of these two SNPs with systemic inflammation is not known, possible explanations point to an increased level of tHcy, which is correlated with serum C4, CRP, and the IgM level [40] as well as increased oxidative stress [41,42] and DNA hypomethylation [43]. These explanations mainly refer to c.665TT homozygotes and c.665CT/c.1298AC double heterozygotes [39].

TNF, as the first released cytokine during injuries or stress [44], is secreted by immune cells and plays multiple roles in immunity, inflammation, homeostasis, cell proliferation, and programmed cell death [45]. MTHFR enzyme polymorphisms may unbalance one-carbon metabolism and affect homocysteine levels, DNA methylation, and the cellular redox state; this could have effects on the RA inflammatory scenario in which TNF plays a key role. Therefore, observing the allele dose-dependent response to anti-TNF-α drugs in the c.665C>T polymorphism could be explained by the level of the MTHFR enzyme function and the degree of the consequences of its dysfunctionality in contributing to inflammation, which could be proportional to the level of TNF. We also observed an allele dose-dependent response, although in an opposite direction, to anti-TNF-α drugs in the c.1298A>C polymorphism, which could be explained by the fact that these two polymorphisms have a strong linkage disequilibrium [46]; the T-C haplotype defined by their minor alleles is very rare [34]. In our study, all c.1298CC homozygotes were also c.665CC wild-type homozygotes. Therefore, considering the influence of the c.1298A>C SNP only on the regulatory domain of the enzyme, patients with the CC homozygous genotype for c.1298A>C had a more functional enzyme compared with the patients with TT homozygous for the c.665C>T variant and were probably less affected by an MTHFR dysfunctionality-related inflammatory process.

TNF-α is initially produced in a precursor transmembrane (tmTNF) form on the cell surface; after cleavage by metalloproteinases, it is then released in a mature and soluble (sTNF) form, which mediates its biological activities through TNF-α receptors type 1 and type 2 (TNF-R1 and TNF-R2) [33]. The five common anti-TNF-α drugs are not structurally identical. Adalimumab and golimumab are full human IgG1 mAbs. Infliximab is a chimeric IgG1 mAb. Certolizumab is a PEGylated Fab fragment of IgG1 mAb without an Fc portion. Etanercept is a dimeric fusion protein of TNF-R2 with IgG1 Fc [33]. All of these anti-TNF-α drugs mainly target and neutralize sTNF; however, activities against tmTNF and Fc receptor-expressing cells have been reported [47,48]. In our study, we observed that after stratifying the cases according to the type of anti-TNF-α drug used in the therapy (i.e., mAbs vs. the fusion protein of the TNF receptor), the trend of the response for the c.1298A>C variant was dissimilar to the c.665C>T SNP, with a significant difference between the two groups (*p* = 0.032). It is known that the distribution, pharmacological half-life, and degradation of anti-TNF-α drugs have several differences [49,50,51]. It has been found that etanercept, as an FP, weakly binds to tmTNF compared with mAbs counterparts [48,52]; unlike mAbs agents, only one etanercept can bind to each molecule of tmTNF, with less efficacy in blocking the downstream activities of tmTNF [53,54]. Considering the observation of an opposite trend of the response to mAb and FP anti-TNF-α drugs for the c.1298AC heterozygotes, it is worth noting that, regardless of the differences in the drug design, it seemed that the efficacy of etanercept (as an FP drug) in comparison with the mAbs agents was being influenced by the c.1298A>C polymorphism, unlike c.665C>T, in the *MTHFR* gene. If different polymorphisms and genotypes of *MTHFR* change the enzyme activity differently, it is possible to expect dissimilar consequences such as different contributions to an inflammatory status, which could be inferred by how slightly different drugs can act differently.

This study should be considered to be preliminary, and the results should be carefully interpreted as it was based on a small cohort of patients. In order to have an established concept of the effects of different polymorphisms on the *MTHFR* gene and their association with drug ADME (absorption, distribution, metabolism, and excretion) as well as the response to different anti-TNF-α therapies in RA patients, a replication study with a larger sample size should be carried out. In addition, considering the strong linkage disequilibrium of these two SNPs, a diplotype analysis is of utmost importance; however, due to the limited number of cases, it was not possible to perform such an analysis.

In conclusion, considering the failure of a phase I therapy in RA patients who have to initiate an anti-TNF-α regime as a phase II therapy, having efficient response-predictive markers such as pharmacogenetic markers could help patients know ahead of time whether the drug is likely to benefit and be safe for them without facing the complications associated with anti-TNF-α drugs. Our study showed that the c.665C>T and c.1298A>C polymorphisms in the *MTHFR* gene influenced the response to the anti-TNF-α therapy in RA patients, although in opposite directions and with a few differences depending on the drug type. Our results suggest a role of one-carbon metabolism in the efficacy of anti-TNF-α drugs, and may pave the way for more personalized interventions in the treatment of RA. In perspective, the development of an accurate prediction model based on a wide range of pharmacogenetic, metabolic [55], clinical [56], circulating microRNA [57,58], and serological [59] markers—which may influence the treatment response with an additive score significance—could tailor better personalized anti-TNF-α treatments for RA patients.

## 4. Materials and Methods

### 4.1. Subjects

A total of 528 RA patients included in the regional RA registry of the Emilia-Romagna region were enrolled from the Rheumatology units of five Italian hospitals, including the Sant’Anna University Hospital, Ferrara; the Rizzoli Orthopaedic Institute, Bologna; the S. Maria Nuova Hospital, Reggio Emilia; the Sant’Orsola-Malpighi University Hospital, Bologna; and the University Hospital, Modena. RA was diagnosed according to the ACR and EULAR criteria of 2010 [60]. A peripheral blood sample was collected from each case before initiating the therapy. For each patient, the anthropometric data (including age, sex, exposure data to tobacco smoking, and serological data for the presence of rheumatoid factor (RF) and anti-citrullinated protein antibodies (APCA)) were collected. Patients were followed-up for 12 months, and the response to the therapy was assessed according to the EULAR response criteria [61,62] and categorized as no response (NR), moderate response (MR), and good response (GR) (Table 3). Written informed consent was obtained from each patient, and the study was reviewed and approved by the ethical board of the Province of Ferrara (16 October 2014).

### 4.2. DNA Extraction and Genotyping

Genomic DNA was extracted from peripheral blood samples using Nucleon BACC1 (GE Healthcare, Chalfont Saint Giles, UK) or QIAamp DNA Blood Mini (Qiagen, Hilden, Germany) kits, according to the manufacturer’s instructions. The DNA was titrated by a Qubit 2.0 fluorometer (Invitrogen, Singapore) using a Qubit dsDNA BR assay kit (Life Technologies, Carlsbad, CA, USA). The genotyping of the two SNPs in the *MTHFR* gene (including c.665C>T (rs1801133) and c.1298A>C (rs1801131)) was performed using pre-designed TaqMan 5′ exonuclease assays (Applied Biosystems, Foster City, CA, USA); subsequently, the fluorescence signals of the probes were detected by an ABI 7300 Real-Time PCR System (Applied Biosystems), according to the supplier’s protocol.

### 4.3. Statistical Analysis

The data were assessed for allele and genotype frequencies and the Hardy–Weinberg equilibrium. The odds ratio (OR) with a 95% confidence interval (CI) was calculated for four different comparisons, including the overall response vs. no response (GR+MR vs. NR), good response vs. no response (GR vs. NR), moderate response vs. no response (MR vs. NR), and good response vs. moderate response (GR vs. MR), assuming a co-dominant genetic model. The analyses were further assessed under dominant and recessive models. A *p*-value < 0.05 was considered to be statistically significant. All analyses were performed using Microsoft Excel (2016).

## Figures and Tables

**Figure 1 ijms-24-04110-f001:**
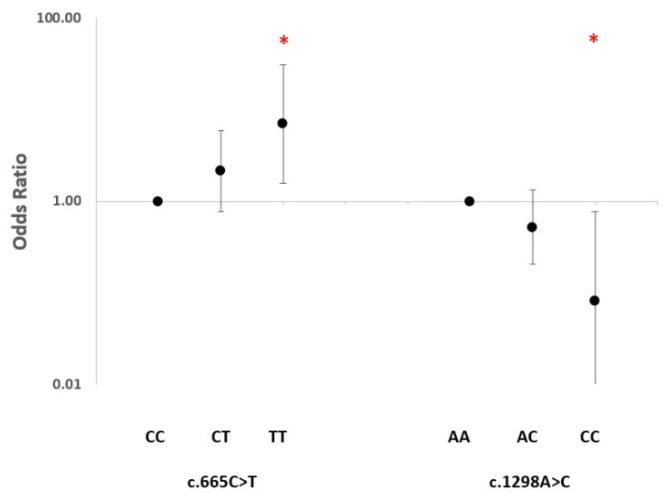
Odds ratio of association of *MTHFR* c.665C>T and c.1298A>C genotypes with response to anti-TNF-α therapy (GR+MR vs. NR) under a co-dominant model and considering wild-type homozygotes as a reference. * Significant nominal *p*-value.

**Figure 2 ijms-24-04110-f002:**
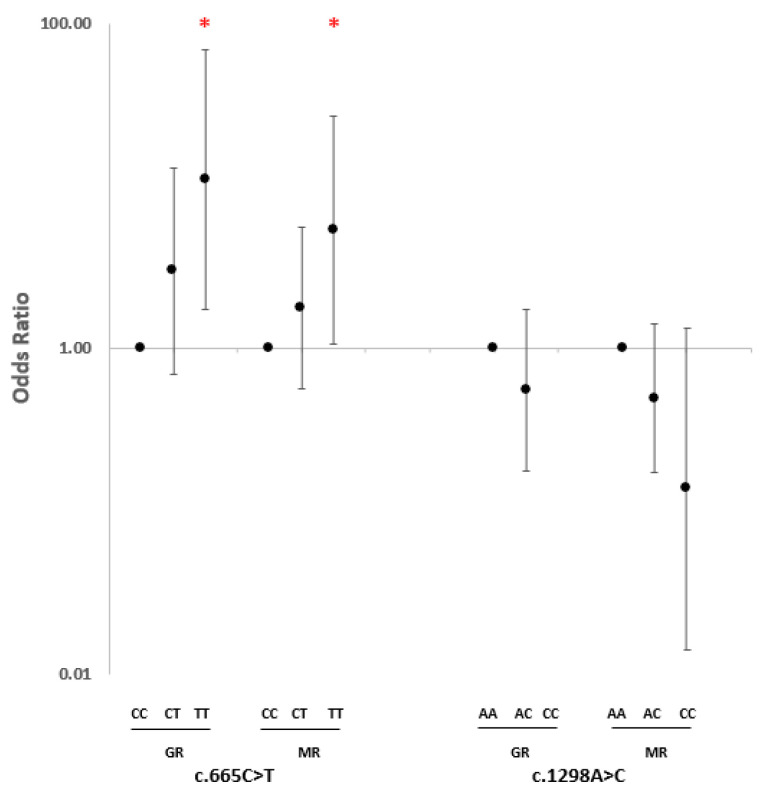
Odds ratio of association of *MTHFR* c.665C>T and c.1298A>C polymorphisms with the level of response to anti-TNF-α therapy considering wild-type homozygotes as a reference. * Significant nominal *p*-value. GR: good response; MR: moderate response.

**Figure 3 ijms-24-04110-f003:**
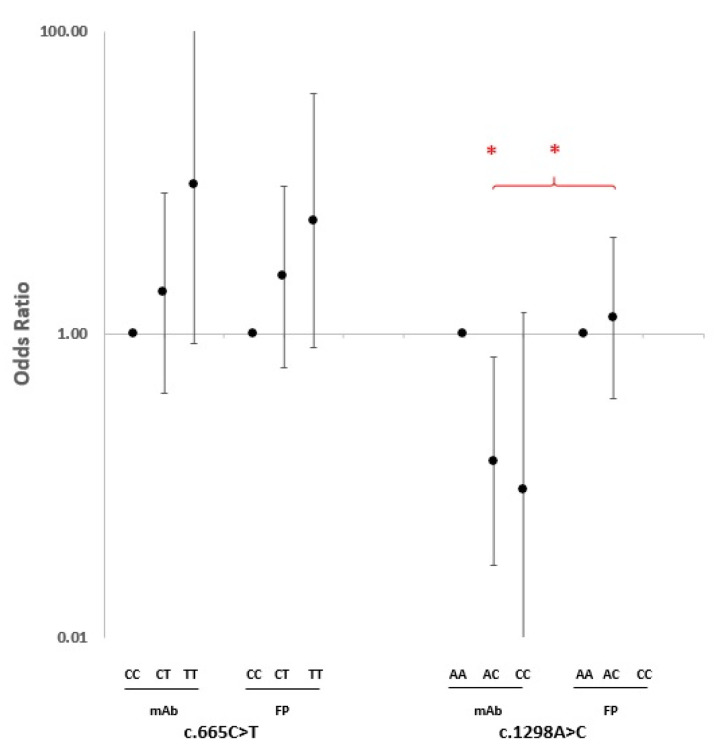
Odds ratio of association of *MTHFR* c.665C>T and c.1298A>C genotypes with response to anti-TNF-α drug subtypes considering wild-type homozygotes as a reference. * Significant nominal *p*-value. mAb: monoclonal antibody; FP: fusion protein.

**Table 1 ijms-24-04110-t001:** Characteristics of the patients included in the study.

Variable (n)	Stratum	(%)
Age (81) *	>60 years	40.74
Sex (81)	Female	67.90
Tobacco smoking (81)	Current smokers	24.69
RF (61)	Positive	78.69
ACPA (59)	Positive	74.58
**Response (85)**		
GR (21)		24.71
MR (30)		35.29
NR (34)		40

* Mean = 56.24 years; SD = 12.96 years; CV = 23%.

**Table 2 ijms-24-04110-t002:** *MTHFR* c.665C>T and c.1298A>C genotype frequencies of the patients included in the study.

SNP (n)	Genotype	Frequency (%)	Allele	Frequency (%)
c.665C>T (81)	CC CT TT	28 51 21	C T	54 46
c.1298A>C (81)	AA AC CC	46 48 6	A C	70 30

**Table 3 ijms-24-04110-t003:** EULAR therapy response criteria using DAS28.

Present DAS28	DAS28 Improvement		
	>1.2	>0.6 and ≤1.2	≤0.6
≤3.2	Good response	Moderate response	No response
>3.2 and ≤5.1	Moderate response	Moderate response	No response
>5.1	Moderate response	No response	No response

## Data Availability

The datasets generated during the study are available from the corresponding author upon reasonable request.

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
