# Peer review of "MTHFR c.665C>T and c.1298A>C Polymorphisms in Tailoring Personalized Anti-TNF-α Therapy for Rheumatoid Arthritis"

_ijms, 2023, doi:10.3390/ijms24044110_

Round 1
Reviewer 1 Report
In this study the authors investigated the association of two RA-related genetic polymorphisms, c.665C>T - historically referred to as C677T - and c.1298A>C, in MTHFR gene as response markers to anti-TNF-α therapy. They found that genetic polymorphisms in the MTHFR gene are associated with response to anti-TNF-α therapy, with potential significance for anti-TNF-α drug type. The topic is interesting. Some concerns and suggestions are listed as below:
Why these two SNPs were selected in this study? How about other SNPs? Previous related studies (or in other disease conditions) should be mentioned in the part of introduction. For example, some polymorphisms in genes TNF, TNF receptor superfamily 1B (TNFR1B) and TNFα-induced protein 3 gene (TNFAIP3) have been associated with response to anti-TNF therapy in patients with psoriasis (PMID: 23337970).
In this study a total of 81 patients who received anti-TNF-α therapy were included. It is not clear for readers if these patients had received different treatments previously (monotherapy or in combination with other drugs?). Choice of therapy may influence drug response.
Apart from genetic polymorphisms, other factors influencing treatment response should be considered. To assess the utility of genetic predictors of response, all other clinical predictors also need to be acknowledged and understood.
In tables 1 and 2, overlapping numbers and texts were noted. Please revise.
In line 113, 'show' should be 'showed'. Please double check the whole manuscript.
I wonder if MTHFR gene has any other effects influencing inflammation or disease.
Regarding genetic polymorphisms, I wonder if these patients have different levels of TNF-α synthesis.
Potential side effects including increased malignancies and infections, or increased congestive heart failure should be provided or discussed.
External validation of genetic biomarkers is required.
The development of anti-drug antibodies (immunogenicity) should not be ignored.
Reviewer 2 Report
The authors aimed to investigate the association of these two SNPs with response to the TNF-α inhibitor drugs in RA patients.
The study covers some issues that have been overlooked in other similar topics. The structure of the manuscript appears adequate and well divided in the sections. Moreover, the study is easy to follow, but some issues should be improved. Some of the comments that would improve the overall quality of the study are:
I-) Authors must pay attention to the technical terms acronyms they used in the text
II-) Please better stated the limitation of the study
Reviewer 3 Report
Dear Authors,
Thank you for your work.
Results:
Table 1 and Table 2, please: insert Mean, SD and CV.
Statistical Analysis:
IBM SPSS? Not used? Why? Please, justify.
References, introduction and discussion:
Please, insert two studies (2022).
Discussion:
Please, insert limitations, practical implications and future research.
Kind regards
Round 2
Reviewer 1 Report
The authors have addressed my concerns.
Reviewer 3 Report
Thanks
Kind regards